# An Evaluation of CRA40 and ERA5 Precipitation Products over China

**Zelan Zhou** [1,2,†], **Sheng Chen** [3,4,†], **Zhi Li** [1,2,*] **and Yongming Luo** [5]

1   Key Laboratory of Environment Change and Resources Use in Beibu Gulf, Nanning Normal University, Ministry of Education, Nanning 530001, China; zhouzelan@email.nnnu.edu.cn

2   School of Geographic Sciences and Planning, Nanning Normal University, Nanning 530001, China

3   Heihe Remote Sensing Experimental Research Station, Northwest Institute of Eco-Environment and Resources, Chinese Academy of Sciences, Lanzhou 730000, China; chensheng@nieer.ac.cn

4   Southern Marine Science and Engineering Guangdong Laboratory (Zhuhai), Zhuhai 519082, China

5   Guangxi Institute of Meteorological Sciences, Nanning 530022, China; mingyongluo858@163.com

*   Correspondence: lizhi@nnnu.edu.cn; Tel.: +86-0771-3908585

†   These authors contributed equally to this work.

**Abstract:** Precipitation datasets derived from reanalysis products play a crucial role in weather forecasting and hydrological applications. This paper aims to evaluate the performance of two distinct reanalysis precipitation products, i.e., the first-generation Chinese global land-surface reanalysis precipitation product (CRA40) and the fifth-generation European reanalysis precipitation product (ERA5), over mainland China. The evaluation is based on continuous and categorical statistical indicators with daily-scale gridded-point rain gauge data obtained from Chinese surface meteorological stations. The findings of this study can be summarized as follows: (1) CRA40 demonstrates a clear superiority over ERA5 in terms of the 13-year daily mean precipitation and seasonal daily precipitation. CRA40 exhibits better correlation coefficients (0.97), relative biases (5.25%), root mean square errors (0.34 mm), and fractional standard errors (0.05). (2) Both reanalyzed precipitation products generally exhibit an overestimation of precipitation in mainland China. The degree of overestimation is particularly pronounced in dry climatic regions (e.g., QZ, XJ), while wet regions (e.g., CJ, HN) demonstrate relatively less overestimation. (3) ERA5 shows better performance in the detection of daily precipitation. Neither CRA40 nor ERA5 can capture heavy precipitation events well. These findings are expected to advance our understanding of the strengths and limitations of the reanalysis precipitation products, CRA40 and ERA5, over China.

**Keywords:** CRA40; ERA5; Chinese mainland; test evaluation; applicability





## 1. Introduction

Precipitation is an important component of the Earth's water cycle, and as a major input for hydrological, meteorological, and climatic modeling, it holds great significance for hydrological, meteorological, and climatological studies [1,2]. Since precipitation exhibits significant variability in both space and time [3], the collection of timely and reliable precipitation data is essential and challenging.

Currently, the main methods of measuring surface precipitation include rain gauges, weather radar, and remote sensing satellites [4,5]. Rain gauges provide a direct physical record of precipitation at specific locations, and the amount of precipitation measured by rain gauges is typically considered as the true value. Weather radar provides continuous coverage of precipitation observations with a high spatial and temporal resolution over an area of hundreds of kilometers. Satellites provide a much larger coverage of precipitation observations than weather radar but have a lower spatial and temporal resolution than weather radar. Satellite-based quantitative precipitation estimation (QPE) products are not limited by topographic factors, while avoiding the high maintenance costs associated with

weather radar systems. Although the aforementioned three methods have been able to meet the requirements for monitoring regional surface precipitation, they still possess certain limitations. Rain gauges suffer from uneven distribution across the ground and small sampling areas, thereby hindering their applicability to large regions and global scales. Weather radar, despite its advantage in precipitation inversion, is susceptible to various sources of error and extreme weather conditions. Furthermore, it faces challenges related to limited spatial coverage in remote areas. Additionally, the high costs associated with the utilization and maintenance of weather radar restrict its application, particularly in developing countries [6]. The precipitation products obtained through the inversion of satellite observations also encounter numerous errors, with common sources of error including disparities in satellite revisit periods, subtle distinctions between remotely sensed signals and precipitation rates, and atmospheric effects [7]. With the development of technology, reanalysis products based on a variety of observations and assimilation techniques have become an alternative method for obtaining estimates of global continuous precipitation.

In the late 1980s, scientists began utilizing data assimilation technology to perform quality control and integration of observational data, such as ground observations, remote sensing satellites, and weather radars, with numerical forecasting products [8,9]. This approach resulted in the generation of gridded datasets with extensive temporal coverage, a broad spatial extent, dynamic and physical significance, and a high spatial and temporal resolution. These datasets are commonly referred to as "reanalysis data". One of the significant advantages of reanalysis products is their ability to provide vertical atmospheric information, encompassing various atmospheric variables aside from precipitation, such as temperature, humidity, and moisture content, at different vertical levels [10,11]. Since the 1990s, global reanalysis products have been released by institutions in the United States, Europe, and Japan. The National Centers for Environmental Prediction (NCEP) and the Center for Atmospheric Research (NCAR) in the United States have developed the NCEP/NCAR reanalysis product [12]. The European Centre for Medium-Range Numerical Forecasts (ECMWF) has produced the reanalysis products ERA40 [13] and ERA-Interim [14]. The Japan Meteorological Agency (JMA) and the Central Research Institute of the Electronics and Energy Industry (CRIEPI) have developed the reanalysis product JRA25 [15] and the upgraded JRA55 [16]. The continuous enhancement and advancement of assimilation technology and weather models have contributed to the improvement and upgrading of reanalysis products. One of the latest and most advanced reanalysis products available today is the European Reanalysis Product (ERA5) [17], which was completely released by the European Centre for Medium-Range Weather Forecasts (ECMWF) in 2019. ERA5 outperforms its previous four generations in terms of its assimilation system sophistication, parameter optimization scheme, spatiotemporal resolution, and the number of output parameters. These advancements have significantly enhanced the evaluation capabilities of ERA5.

Compared to other countries, China's development of atmospheric reanalysis products started relatively late. In November 2013, the China Meteorological Administration (CMA) initiated the China Global Atmospheric Reanalysis Program (CGARP). In May 2021, the CMA officially released China's first-generation atmospheric reanalysis product (CRA), aiming to enhance the accuracy of historical weather analyses and facilitate in-depth investigations of various weather and climate systems. This product provides a global representation of three-dimensional atmospheric conditions from 1979 to 2018, spanning from the surface to an altitude of 55 km. It offers a temporal resolution of 6 h and a horizontal resolution of 34 km. The availability of this product fills the gap in China's global atmospheric reanalysis field. In addition, the atmospheric reanalysis product (CRA) consists of two types of products: the global atmospheric reanalysis product (CRA-40) and the global land surface reanalysis product (CRA-40/Land). Distinguishing itself from other reanalysis products, the CRA incorporates a wide range of high-quality data sources. These include bias-corrected sounding instruments [18], ground-based observations on the Tibetan Plateau, integrated aircraft datasets [19], recently released AMW reprocessed observations, and GPS ocean observations [20]. The data quality assessment of CRA-40

products indicates that they are generally on par with the international third-generation global reanalysis products [21].

　　Precipitation datasets derived from reanalysis products are highly valuable for weather forecasting and hydrological applications [22]. However, it is important to recognize that reanalysis products do not represent actual observations. Instead, they are a combination of numerical forecasts and observational data. Consequently, uncertainties arise from various sources, including numerical models, assimilation schemes, homogenization processes, and errors resulting from changes in the observation system [23]. Therefore, it becomes crucial to evaluate the quality and suitability of reanalysis precipitation products. To date, both domestic and international researchers have conducted extensive assessments on the applicability of monthly-scale and hourly-scale reanalysis precipitation products in the Chinese region. Wu et al. [24] conducted a comparative analysis of spatial variations in extreme precipitation and temperature indices using ERA-Interim and NCEP/DOE reanalysis data, based on station observations in the Jianghuai Basin from 1979 to 2011. Xin et al. [25] conducted an evaluation of the applicability of two reanalysis precipitation products, ERA5-Land and ERA5-HRES, in the Guangdong–Hong Kong–Macao Greater Bay Area of China. This study revealed that the ERA5 products predominantly captured moderate-intensity precipitation, demonstrating their strongest performance during the rainy season in the southern coastal areas of the Greater Bay Area. Moreover, the ERA products exhibited notable accuracy in the northern mountainous areas and vegetation zones during the dry season. Based on the data from 807 meteorological stations in mainland China during the period of 2001–2017, Jiang et al. [26] evaluated the performances of ERA5, the Integrated Multi-Satellite Retrieval of Global Precipitation Measurements (IMERG), Tropical Precipitation Measurement Mission Multi-Satellite Precipitation Analysis (TMPA), and the Climate Prediction Center (CPC) precipitation products, and found that ERA5 exhibits large statistical errors. In a study by Qu et al. [27], the suitability of four reanalysis monthly precipitation products, namely CRA-40/Land, ERA5, JRA55, and MERRA2, was assessed in Inner Mongolia using measured precipitation data from stations spanning the period of 1981 to 2020. This study revealed that CRA-40/Land and MERRA2 exhibited higher data quality compared to ERA5, with ERA5 being ranked third in terms of data quality. Wang et al. [9] assessed the applicability of CRA-40/Land, ERA5, and the U.S. reanalysis product CFSR monthly precipitation products in the Chinese mainland region with observed precipitation data from 2231 stations in the Chinese mainland region as a reference. This study found that CRA-40/Land had the smallest error for monthly average precipitation, and demonstrated the highest applicability in China. Li et al. [28] evaluated the precipitation performance of China's Global Atmospheric Reanalysis precipitation data (CRAI) and ECMWF Intermediate Reanalysis (ERAI), Japan's 55-Year Reanalysis (JRA55), and the NCEP Climate Prediction System Reanalysis (CFSR), and compared the results with the observational data. The CRAI, ERAI, and JRA55 tend to overestimate light and moderate precipitation, but underestimate heavy and extreme precipitation. Huang et al. [29] evaluated the applicability of ERA5-Land precipitation data in southwest China based on the observation data of 441 surface rain gauges of the China Meteorological Administration from 2018 to 2020. The results showed that ERA5-Land precipitation was most obviously intensive in Tibet, and the error was relatively large in mountainous areas with complex terrain. Furthermore, Jiang et al. [30] evaluated the accuracy of ERA5-day precipitation data in mainland China from 2003 to 2015. The results indicated that ERA5 exhibited high root-mean-square errors in subtropical climatic zones and humid tropical climates. Additionally, it overestimated light rainfall events and underestimated medium–heavy precipitation events. However, the assessment of the applicability of the CRA-40/Land daily precipitation product over large-scale mainland China has not been widely conducted thus far.

　　The object of this study is to evaluate the applicability of CRA-40/Land daily precipitation products over mainland China. Observations from 2301 surface rain gauge stations in the mainland China region were used as a reference to compare ERA5 daily precipitation

products with CRA40. The paper is organized as follows: Section 2 introduces the study area and data used in this study. Section 3 describes the evaluation results and discussions with an analysis of the spatio-temporal error characteristics. Section 4 gives the summary and conclusions.

## 2. Materials and Methods

### 2.1. Study Area

Mainland China, with its diverse natural and climatic geomorphic units, provides an ideal case study area for evaluating the accuracy of reanalysis precipitation products. In this study, mainland China is utilized as the research domain, which encompasses a latitudinal and longitudinal range of 73°40′–135°02′E and 18°10′–53°33′N, respectively. The distribution of the topography is depicted in Figure 1. Referring to Chen's study [31], eight subregions are defined based on the distribution of mean annual precipitation, elevation, and mountain range orientation, including (I) Xinjiang region (XJ) in far-west China, (II) the high-elevation Qinghai–Tibetan Plateau region (QZ), (III) the northwestern region (XB), (IV) the region north of the Yanshan Mountains in northeastern China (DB), (V) the northern part of China, north of the Huaihe River and north of the Qinling Mountains (HB), (VI) the southwestern Yunnan–Guizhou Plateau region (YG), (VII) the Yangtze River Plain region (CJ), and (VIII) the southeastern region of China (HN).

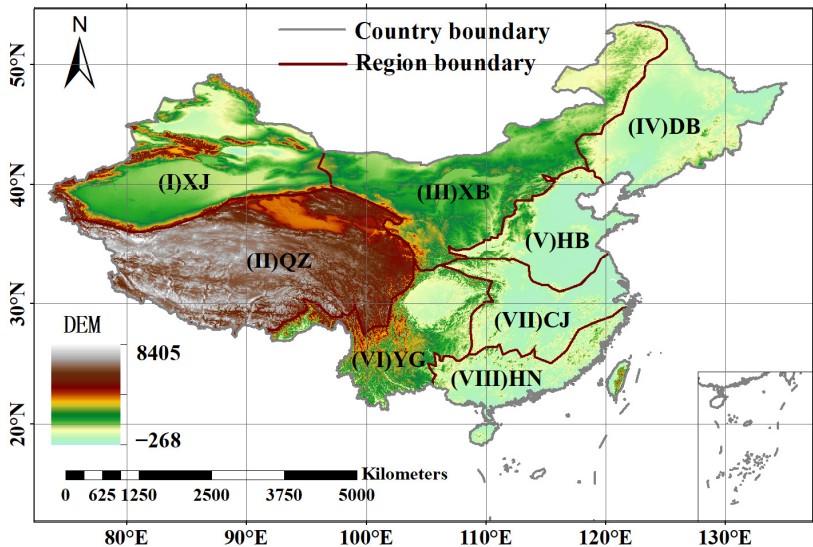

**Figure 1.** Topographic features and subdivisions of China (I–VIII in order of Xinjiang, Qinghai–Tibetan Plateau, Northwest, Northeast, North China, Western Yunnan–Guizhou Plateau, Middle and Lower reaches of Yangtze River, Southeast).

Mainland China is geographically situated in the eastern part of Eurasia and on the west coast of the Pacific Ocean, covering a large extent of longitude and latitude. The spatial distribution of precipitation in mainland China displays significant variability, owing to the external geographic characteristics and the complex and varied internal terrain and topography. The average annual precipitation demonstrates a decreasing trend from the southeastern coast to the northwestern interior. Furthermore, the seasonal variation in precipitation in mainland China is pronounced due to the influence of monsoon activities, characterized by more precipitation during summer and less during winter.

### 2.2. Ground Reference Data

The surface reference precipitation data utilized in this study were obtained from the daily rain gauge observations (including rainfall and snowfall) collected from 2003 to 2015 by the China Meteorological Information Center (https://data.cma.cn/, accessed on 10 December 2021). Figure 2 exhibits the distribution of 2301 stations across the study

area. The collected surface rain gauge observations were subjected to rigorous quality control procedures and subsequently interpolated into a regular gridded analysis (hereafter referred to as "GGKRIG") at a resolution of 0.25° using the Kriging interpolation algorithm. The Kriging interpolation method is a geostatistical-based interpolation technique that integrates the relevant sample data and considers the spatial structure characteristics within a specific range of unsampled points by assigning individual weight coefficients to each sample value. This results in an unbiased, linear, and optimal estimation value as well as the corresponding estimation variance [32]. However, it should be noted that the use of the algorithm over a relatively large study area with an uneven distribution of ground stations may introduce some uncertainty in the precipitation estimates and may not adequately represent precipitation over the entire region [33,34]. Moreover, in many previous evaluation studies, there have been precedents using the longitude and latitude coordinates of rain gauge stations to extract grid precipitation products for comparative evaluation [35,36]. To minimize the impact of the interpolation on the accuracy and error assessment of the reanalysis products, station observations data has been selectively employed for comparative analysis in this paper. The GGKRIG was only used in the comparison of the spatial distribution of the 13-year daily mean precipitation and seasonal daily mean precipitation to capture the overall trend of the spatial distribution of precipitation.

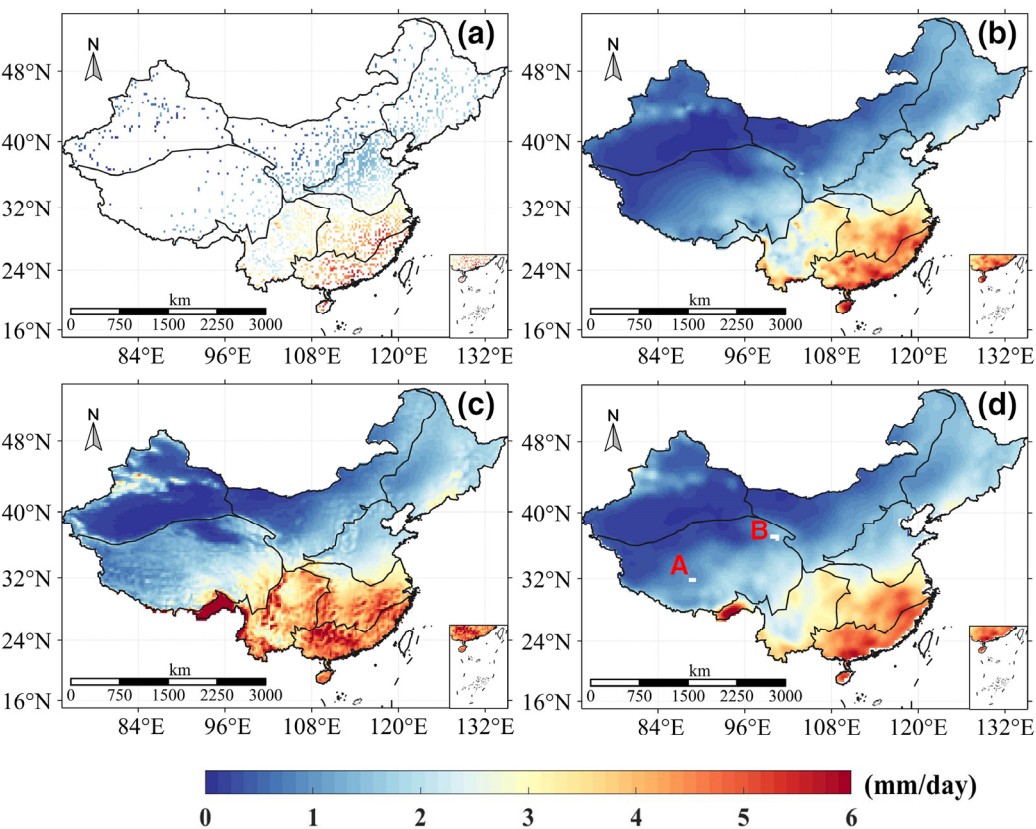

**Figure 2.** Spatial distribution of 13-year daily evaluated precipitation for (**a**) Gauge; (**b**) GGKRIG; (**c**) ERA5; and (**d**) CRA40.

### 2.3. CRA40

The CMA has developed the first generation of global reanalysis products known as the CRA (CMA's global atmospheric reanalysis). This reanalysis incorporates various conventional ground-based observations, including China Fundamental Data, Climate Forecasting System Reanalysis (CFSR), and the Integrated Surface Database (ISD). The CRA consists of two components: global atmospheric reanalysis products (CRA-40) and global land surface reanalysis products (CRA-40/Land). These products provide 3-hourly,

daily, and monthly data with a spatial resolution of approximately 34 km. The land surface reanalysis products encompass near-surface atmospheric drive analysis and land surface model simulation. The land surface model simulation products are generated by driving the Noah 3.3 land surface model with near-surface atmospheric forcing analysis products. Specifically, this study focuses on the day-by-day precipitation products (including precipitation and snow) simulated using the CRA40-40/Land land-surface model (referred to as CRA40) over a 13-year period from January 2003 to December 2015. The spatial resolution of the CRA40 precipitation product is $0.315° \times 0.315°$ ($1152 \times 576$, Gaussian grid). To facilitate subsequent calculations and comparisons, the spatial resolution of CRA40 was resampled to $0.25° \times 0.25°$ using bilinear interpolation.

It is important to note that the CRA40 precipitation product lacks sea surface precipitation information, resulting in data gaps in coastal areas. Moreover, there are also missing data for some grids within China, the number of missing grids increases further after bilinear interpolation, and the missing grid areas are indicated by the red letters A and B in Figure 2d. In order to mitigate the impact of missing CRA40 values on the assessment results, 1950 station observations (referred to as the gauges) containing valid CRA40 values were selected from 2301 rain gauge station observations as the reference data for subsequent calculations and analyses.

## 2.4. ERA5

ERA5 is the latest publicly available atmospheric reanalysis product from the ECMWF, providing a powerful global climate detection dataset. It utilizes the 4D-Var data assimilation system and combines it with observations from around the world, resulting in higher spatial and temporal resolution compared to the previous ERA-Interim product. The atmospheric data can be used at different altitudes, and the precipitation data consists of two 2D surface-level parameters, rainfall and snow. ERA5 generates large-scale precipitation using a cloud scenario that is inverted by a convective scenario to convective precipitation.

In this study, the ERA5 precipitation dataset spans from January 2003 to December 2015, with a spatial resolution of $0.25° \times 0.25°$ and a temporal resolution of hours through the C3S Climate Data Store application programming interface (API) (https://cds.climate.copernicus.eu/#!/home, accessed on 7 October 2022). To facilitate subsequent processing and analysis, hourly precipitation was accumulated to daily precipitation to match CRA40 for subsequent processing and analysis.

## 2.5. Evaluation Methods and Metrics

In this paper, eight metrics were chosen to provide a comprehensive assessment of accuracy and error for reanalysis precipitation products. These eight metrics can be categorized into two groups [37]. The first group consists of continuous statistical indicators, including root-mean-squared error (RMSE), bias, relative bias (RB), correlation coefficient (CC), and fractional standard error (FSE), which quantify the accuracy of precipitation data. The RMSE provides an overall measure of accuracy and error level for precipitation products. RB indicates the deviation of precipitation products from ground observation data, with negative and positive values denoting underestimation and overestimation, respectively. The CC reflects the linear correlation between precipitation products and ground observation data. FSE captures the difference between the mean values of precipitation products and ground observation data, with smaller values indicating higher accuracy. The second category comprises categorical statistics, namely the probability of detection (POD), the false alarm ratio (FAR), and the critical success index (CSI). The POD describes the proportion of precipitation events detected with the reanalysis precipitation products relative to the proportion of ground-observed precipitation events. The FAR represents the proportion of precipitation events with false alarms from the precipitation products among the total number of detected precipitation events. The CSI is a comprehensive measure that considers both the POD and the FAR, representing the ability of precipitation products to detect real precipitation events. These indicators have been computed

on a pixel-by-pixel basis over China, and the specific formulas and explanations can be found in Table 1. Through comparative analysis, a more comprehensive understanding of the detection ability of the CRA40 and ERA5 products on mainland China precipitation can be achieved.

**Table 1.** Calculation formula for assessment indicators.

| Evaluation Indicators | Formula | Range of Values | Optimal Value | Unit |
|---|---|---|---|---|
| Bias | $E_i - G_i$ | $(-\infty, \infty)$ | 0 | mm/day |
| RMSE | $\sqrt{\frac{1}{n}\sum_{i=1}^{n}(E_i - G_i)^2}$ | $(-\infty, \infty)$ | 0 | mm/day |
| RB | $\frac{\sum_{i=1}^{n}(E_i - G_i)}{\sum_{i=1}^{n} G_i} \times 100\%$ | $[0, \infty)$ | 0 | % |
| CC | $\frac{\sum_{i=1}^{n}(E_i - \overline{E})(G_i - \overline{G})}{\sqrt{\sum_{i=1}^{n}(E_i - \overline{E})^2}\sqrt{\sum_{i=1}^{n}(G_i - \overline{G})^2}}$ | $[-1, 1]$ | 1 | / |
| FSE | $\sqrt{\frac{\frac{1}{n}\sum_{i=1}^{n}(E_i - G_i)^2}{Avg(G_i)}}$ | $[0, \infty)$ | 0 | / |
| POD | $\frac{Hits}{Hits + Miss}$ | $[0, 1]$ | 1 | / |
| FAR | $\frac{False\ Alarm}{Hits + False\ Alarm}$ | $[0, 1]$ | 0 | / |
| CSI | $\frac{Hits}{Hits + Miss + False\ Alarm}$ | $[0, 1]$ | 1 | / |

Note: $n$ is the total number of grid points; $E_i$ and $G_i$ denote the reanalysis precipitation products and the interpolated precipitation with the rainfall gauge observations, respectively; $\overline{E}$ is the mean value of the reanalysis precipitation products; $\overline{G}$ is the average precipitation value interpolated with the ground rainfall gauge observations; $Avg(G_i)$ is the average of the values interpolated to the grid points based on the rainfall gauge observations. Hits is the number of precipitation events detected by both reanalysis precipitation products and gauge-observed precipitation; Miss is the number of events that reanalysis precipitation products show no precipitation but gauge observations show precipitation; False Alarm is the number of events that reanalysis precipitation products show precipitation but gauge observations show no precipitation.

For further quantitative comparison, precipitation can be divided into six categories according to the Chinese national standard "Classification of Precipitation Levels" (GB/T 28592-2012) [38], as shown in Table 2.

**Table 2.** Daily precipitation test classification standard.

| Class | Intensity (mm/day) | Rank |
|---|---|---|
| 1 | 1~10 | Light Rain |
| 2 | 10~25 | Medium Rain |
| 3 | 25~50 | Heavy Rain |
| 4 | 50~100 | Rainstorm |
| 5 | 100~250 | Large Rainstorm |
| 6 | ≥250 | Extreme Rainstorm |

Note: the minimum amount of light rain is adjusted to 1 mm/day to avoid the effects of sporadic precipitation [39].

## 3. Results

### 3.1. 13-Year Daily Mean Precipitation

Figure 2 illustrates the spatial distribution of the 13-year mean daily precipitation for the gauges, GGKRIG, and ERA5 and CRA40 in mainland China from 2003 to 2015. As depicted in Figure 2a, the rain gauge stations are densely located in southeastern China while being sparser in the northwestern region. Spatially, both ERA5 and CRA40 (Figure 2b–d) generally capture the spatial variation trend in China, which align with the gauge-based analysis, GGKRIG. This trend manifests as a decrease in precipitation from the southeastern coastal areas to the northwestern inland areas of mainland China. The precipitation-rich regions indicated with CRA40 and ERA5 primarily occur in southeastern China and the southern Himalayan Mountains, with reduced rainfall in the northern and northwestern inland areas exhibiting lower precipitation levels. Nonetheless, noticeable

discrepancies exist between the two products in certain regions. Across China, ERA5 tends to overestimate precipitation in the semi-arid zone near the Tien Shan mountain range in XJ, the southeastern part of QZ, the northwestern part of HN, and the southern part of YG, as well as the central region and the surroundings of the Sichuan Basin. On the other hand, CRA40 demonstrates a spatial precipitation distribution more similar to that of the GGKRIG and superior to ERA5.

Figure 3 shows the scatter density plots of the ERA5 product and the CRA40 product against the gauges to quantitatively compare the 13-year daily average precipitation in mainland China from 2003 to 2015. It should be kept in mind that only the grids that are overlapped with at least one rainfall gauge are selected for comparison in the scatter density plots. It is noted that CRA40 performs better than ERA5 in terms of RB, the RMSE, CC, and FSE of the annual daily average precipitation. It is worth noting that in terms of the CC, the correlation between the CRA40 product and the real data gauges reaches 0.97, while the correlation of the ERA5 product is relatively slightly lower at 0.91. In terms of RB, both of them show overestimation, with a large degree of deviation in precipitation for ERA5 (18.59%) and a relatively small degree of deviation for CRA40 (5.25%). Such obvious overestimation of precipitation by ERA5 in some areas can be found in Figure 2c.

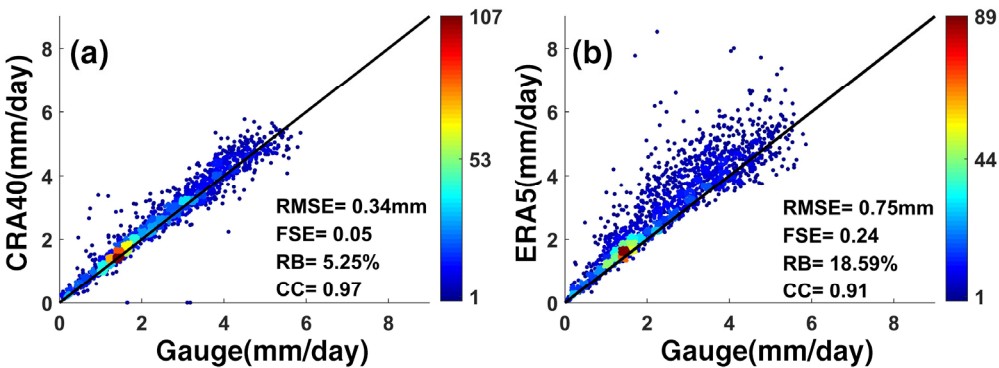

**Figure 3.** The scatter density plot of Gauge versus (**a**) CRA40, (**b**) ERA5.

Meanwhile, in addition to the distributional characteristics of point densities that can be observed in the scatter density plots, a more pronounced difference in the distributional characteristics of precipitation can also be seen. When the points are biased towards the vertical axis, where the reanalysis precipitation products are located, it means that the corresponding reanalysis precipitation product overestimates the precipitation intensity on the ground. In addition, the density of points is marked in color; the closer to the vertical axis, the higher the density of points, and the more the reanalysis precipitation product overestimates the precipitation. The higher density of points of CRA40 and ERA5 are both concentrated in the region composed of the coordinate set of {x, y | x ∈ [0, 3], y ∈ [0, 3]}. Among them, the points of CRA40 are mostly linearly distributed along the line 1:1, indicating that the degree of bias is small and the analyzed surface precipitation is better than that of ERA5, while there are individual lattice points that significantly underestimate the surface precipitation within [1–4) mm/day. The points of ERA5 show a more significant distribution of bias towards the vertical axis, suggesting that they significantly overestimate the intensity of precipitation observed with the surface rainfall gauge. It is noted that some points of ERA5 are high above the 1:1 line when precipitation rates are between 2 and 6 mm/day, implying that ERA5 substantially overestimates precipitation in some regions [40,41].

Table 3 provides more detailed statistics for the eight subregions in China, and the values of the metrics have been rounded to two decimals, with the best performance in different regions and time periods highlighted in bold. As a whole, CRA40 shows better performance compared to ERA5. It is worth noting that CRA40 and ERA5 perform poorly in the arid and semi-arid region of XJ and the higher-altitude region of QZ. In terms of RB, ERA5 significantly overestimates the XJ (49.35%), QZ (70.87%), and YG regions (36.67%),

while CRA40 relatively overestimates XJ (31.40%) and QZ (23.12%) as well as DB (11.04%), and exhibits slight overestimation in all other regions. In terms of the CC, ERA5 has low correlation in the XJ (0.74), QZ (0.74), YG (0.50), and HN (0.4) regions, while CRA40 has poor correlation in XJ (0.68), YG (0.75), and HN (0.69).

**Table 3.** RMSE, RB, FSE, and CC for the 13-year daily average and seasonal daily precipitation by region.

| Indexes | Time | Type | China | XJ | QZ | XB | DB | HB | YG | CJ | HN |
|---|---|---|---|---|---|---|---|---|---|---|---|
| RMSE | 13 years | CRA40 | **0.34** | **0.41** | **0.51** | **0.16** | **0.21** | **0.20** | **0.37** | **0.38** | **0.49** |
| | | ERA5 | 0.75 | 0.50 | 1.21 | 0.43 | 0.29 | 0.30 | 1.30 | 0.61 | 0.86 |
| | Spring | CRA40 | **0.41** | **0.44** | **0.44** | **0.13** | **0.19** | **0.17** | **0.50** | **0.47** | **0.77** |
| | | ERA5 | 0.97 | 0.50 | 0.98 | 0.36 | 0.37 | 0.21 | 1.42 | 1.08 | 1.71 |
| | Summer | CRA40 | **0.75** | **0.88** | **1.18** | **0.37** | 0.49 | **0.50** | **0.81** | **0.78** | **1.14** |
| | | ERA5 | 1.22 | 1.02 | 2.44 | 0.69 | **0.48** | 0.69 | 1.92 | 0.92 | 1.22 |
| | Autumn | CRA40 | **0.29** | **0.29** | **0.41** | **0.15** | **0.15** | **0.14** | **0.34** | **0.33** | **0.47** |
| | | ERA5 | 0.70 | 0.43 | 1.13 | 0.57 | 0.27 | 0.28 | 1.30 | 0.37 | 0.56 |
| | Winter | CRA40 | **0.16** | 0.13 | **0.14** | **0.04** | **0.09** | **0.09** | **0.21** | **0.23** | **0.23** |
| | | ERA5 | 0.52 | **0.17** | 0.41 | 0.19 | 0.15 | 0.12 | 1.02 | 0.45 | 0.61 |
| RB (%) | 13 years | CRA40 | **5.25** | **31.40** | **23.12** | **6.92** | **11.04** | **5.13** | **2.65** | **2.95** | **4.51** |
| | | ERA5 | 18.59 | 49.35 | 70.87 | 28.91 | 16.15 | 8.94 | 36.67 | 8.83 | 8.72 |
| | Spring | CRA40 | **6.89** | **27.58** | **29.34** | **10.46** | **14.37** | **8.39** | **9.51** | **3.24** | **4.42** |
| | | ERA5 | 22.93 | 58.79 | 78.34 | 40.03 | 30.69 | 9.46 | 45.37 | 12.87 | 16.20 |
| | Summer | CRA40 | **4.77** | 52.49 | **21.24** | **6.25** | 8.58 | **4.64** | **0.77** | **3.49** | 2.71 |
| | | ERA5 | 12.81 | **40.59** | 56.05 | 16.76 | **7.53** | 8.47 | 21.94 | 8.65 | **1.15** |
| | Autumn | CRA40 | **3.74** | **13.04** | **20.30** | **5.20** | **11.45** | **1.07** | **−1.44** | **2.55** | 8.75 |
| | | ERA5 | 20.32 | 63.50 | 83.93 | 37.60 | 24.12 | 6.37 | 38.38 | 4.51 | **5.28** |
| | Winter | CRA40 | **6.86** | **−4.94** | **51.62** | **14.36** | **37.61** | **18.64** | **9.93** | **1.16** | **6.57** |
| | | ERA5 | 36.61 | 33.86 | 299.84 | 120.62 | 61.94 | 23.73 | 141.78 | 4.88 | 24.40 |
| FSE | 13 years | CRA40 | **0.05** | **0.36** | **0.19** | **0.02** | **0.03** | **0.02** | **0.05** | **0.04** | **0.05** |
| | | ERA5 | 0.24 | 0.53 | 1.07 | 0.17 | 0.06 | 0.05 | 0.59 | 0.10 | 0.17 |
| | Spring | CRA40 | **0.07** | **0.39** | **0.20** | **0.02** | **0.04** | **0.02** | **0.10** | **0.05** | **0.11** |
| | | ERA5 | 0.40 | 0.51 | 0.98 | 0.18 | 0.13 | 0.04 | 0.83 | 0.25 | 0.53 |
| | Summer | CRA40 | **0.12** | **1.00** | **0.44** | **0.06** | **0.06** | **0.06** | **0.11** | **0.11** | **0.17** |
| | | ERA5 | 0.33 | 1.33 | 1.90 | 0.20 | 0.06 | 0.11 | 0.64 | 0.15 | 0.20 |
| | Autumn | CRA40 | **0.05** | **0.23** | **0.15** | **0.02** | **0.02** | **0.01** | **0.05** | **0.04** | **0.08** |
| | | ERA5 | 0.26 | 0.49 | 1.11 | 0.29 | 0.07 | 0.05 | 0.64 | 0.06 | 0.11 |
| | Winter | CRA40 | **0.04** | **0.08** | **0.19** | **0.01** | **0.04** | **0.02** | **0.07** | **0.03** | **0.03** |
| | | ERA5 | 0.38 | 0.13 | 1.56 | 0.32 | 0.10 | 0.04 | 1.84 | 0.11 | 0.23 |
| CC | 13 years | CRA40 | **0.97** | 0.68 | **0.80** | **0.95** | **0.95** | **0.95** | **0.75** | **0.88** | **0.69** |
| | | ERA5 | 0.91 | **0.74** | 0.74 | 0.90 | 0.93 | 0.88 | 0.50 | 0.80 | 0.40 |
| | Spring | CRA40 | **0.98** | 0.67 | **0.88** | **0.95** | **0.94** | **0.98** | **0.88** | **0.97** | **0.89** |
| | | ERA5 | 0.94 | **0.79** | 0.82 | 0.93 | 0.91 | 0.93 | 0.81 | 0.89 | 0.65 |
| | Summer | CRA40 | **0.93** | 0.58 | **0.72** | **0.92** | **0.92** | **0.90** | **0.75** | 0.65 | **0.61** |
| | | ERA5 | 0.88 | **0.67** | 0.61 | 0.87 | 0.91 | 0.85 | 0.57 | **0.66** | 0.59 |
| | Autumn | CRA40 | **0.95** | 0.68 | **0.85** | **0.97** | **0.95** | **0.94** | **0.80** | 0.66 | **0.93** |
| | | ERA5 | 0.84 | **0.75** | 0.83 | 0.90 | 0.94 | 0.88 | 0.43 | **0.75** | 0.83 |
| | Winter | CRA40 | **0.98** | **0.77** | 0.36 | **0.86** | **0.94** | **0.99** | **0.69** | **0.96** | **0.93** |
| | | ERA5 | 0.86 | **0.77** | 0.76 | 0.65 | **0.94** | 0.95 | 0.54 | 0.82 | 0.71 |

Note: best performance is highlighted in bold.

### 3.2. Seasonal Daily Mean Precipitation

Southeast China has a predominantly monsoon climate, while northwest China has an arid and semi-arid climate. The location of China determines that precipitation in China will show different geographic patterns seasonally, influenced by Asian monsoons, the Tibetan Plateau, and the prevailing westerly winds from the North Atlantic Ocean. Figure 4 shows the spatial distribution of the spatially continuous rain gauge data based on the Kriging interpolation method and the seasonal daily mean precipitation of the CRA40 and ERA5 reanalysis products for mainland China. Figure 5 quantifies the seasonal

performance of the two reanalysis precipitation products using scatter density plots, and Table 3 provides more detailed seasonal statistics for the eight subregions, where the RMSE, FSE, RB, and CC are calculated by comparing the seasonal daily mean precipitation with that of the gauges.

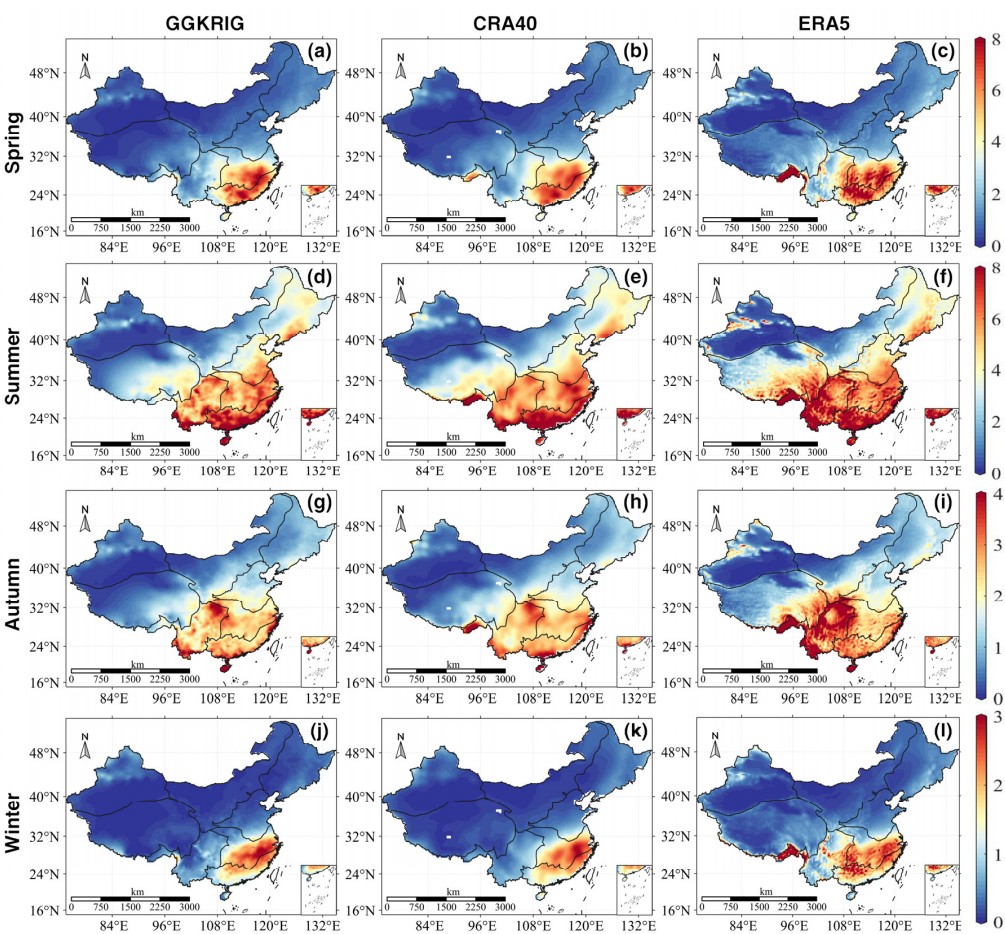

**Figure 4.** Spatial distribution of seasonal daily mean precipitation in (**a**–**c**) spring, (**d**–**f**) summer, (**g**–**i**) autumn and (**j**–**l**) winter for GGKRIG, CRA40 and ERA5 in mainland China, in mm/day.

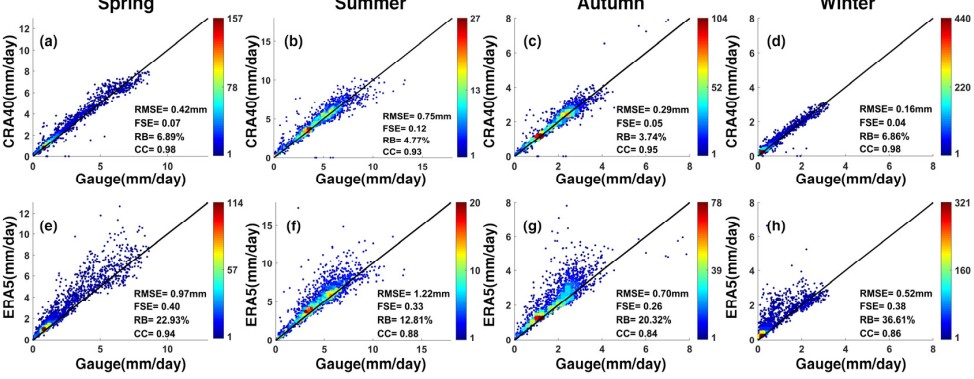

**Figure 5.** (**a**–**d**) Scatterplots of seasonal daily precipitation densities for CRA40 versus Gauge in spring, summer, autumn and winter. (**e**–**h**) Scatterplot of seasonal daily precipitation densities for ERA5 product versus Gauge in spring, summer, autumn, and winter. The colors indicate frequency of occurrence.

As shown in Table 3, both CRA40 and ERA5 tend to overestimate the seasonal daily precipitation in the eight subregions of China. Overall, CRA40 performs better than

ERA5 in assessing seasonal precipitation in most subregions. In the spring season, CRA40 simulates a significant overestimation of precipitation in XJ, QZ, XB, and DB (27.58%, 29.34%, 10.46%, and 14.37%, respectively), while showing relatively better performance in other subregions. On the other hand, ERA5 generally overestimates precipitation in all eight subregions, particularly in XJ, QZ, XB, DB, and YG (58.79%, 78.34%, 40.03%, 30.69%, and 45.37%, respectively). Only in HB does ERA5 show relatively lower overestimation (9.46%), but is still worse than CRA40. In the summer season, both CRA40 and ERA5 exhibit overestimation in all subregions, with greater overestimation in XJ and QZ compared to other regions (CRA40: 52.49%, 21.24%; ERA5: 40.59%, 56.05%). In the fall season, CRA40 demonstrates relatively small RB across regions, with significant overestimation only in XJ, QZ, and DB (13.04%, 20.30%, 11.45%), and slight underestimation in YG (−1.44%). However, ERA5 shows significant overestimation in XJ, QZ, XB, and YG (63.50%, 83.93%, 37.60%, 38.93%). It should be noted that both products perform relatively poorly in assessing winter precipitation compared to the other three seasons. CRA40 underestimates precipitation in XJ (−4.94%) and significantly overestimates it in QZ and DB (51.62%, 37.61%). On the other hand, ERA5 shows significant overestimation in all seven subregions except CJ (≥23.99%), with highly significant overestimation in QZ, XB, and YG (299.84%, 120.62%, and 141.78%, respectively). In general, both CRA40 and ERA5 perform relatively well in assessing seasonal precipitation for HN, CJ, and HB in southeastern China, with CRA40 exhibiting smaller a FSE and higher CC. However, there is room for improvement in their estimations of winter precipitation. In particular, both products show a significant overestimation of precipitation in QZ during winter, and the CCs of CRA40 are smaller in winter in all subregions when compared to the other three seasons.

### 3.3. Daily Precipitation

Figure 6 presents the daily variations in RB, bias, CCs, and RMSE for the two reanalysis precipitation products in mainland China from 2003 to 2015. The daily bias and RMSE generally exhibit annual periodicity and some correlation, with the highest RMSE occurring in summer and the lowest in winter. This pattern is particularly pronounced in the XJ and QZ regions. However, the daily CC values fluctuate dramatically and do not show a clear trend. This indicates that the two reanalysis products are strongly influenced by seasonal climate and surface characteristics. Furthermore, in terms of RB, it can be seen that CRA40 and ERA5 also show a certain regularity, i.e., the maximum value of RB often occurs in the winter, and the minimum value mostly occurs in the summer. This phenomenon is more clearly visible in the Chinese region. Individual dates with very large RB values were found in eight subregions of China, mainly in winter, and CRA40 showed more days with very large RB values. After separately analyzing the single day data with significant RB values, it was found that the reason for the significant RB values was due to the fact that the gauges captured a negligible amount of precipitation on a single day, whereas the reanalysis of the precipitation product simulated significantly more precipitation. In terms of the daily RB, bias, CC, and RMSE, ERA5 generally outperforms CRA40. Although CRA40 exhibits relatively small daily precipitation deviations, the daily CCs are generally significantly lower (Figure 6b,c). In the QZ and YG regions, ERA5 shows a more pronounced positive bias compared to CRA40, especially in summer. The large positive bias of ERA5 in these regions may be the main reason for its poorer performance in summer (Figure 6j,z). It is worth noting that both ERA5 and CRA40 display more negative deviations in the CJ and HN regions, particularly in summer and fall. Both CRA40 and ERA5 tend to assimilate observed and simulated data at larger spatial scales, but they may have limitations in capturing localized precipitation events, resulting in the underestimation of precipitation. Overall, these findings highlight the influence of seasonality, surface characteristics, and regional factors on the performance of the two reanalysis precipitation products in mainland China. In general, the quality of ERA5 products is better than that of CRA40 in daily-scale precipitation analyses in China and in various subregions, but both products have their

limitations in capturing local precipitation events, especially during the rainy season and in complex terrain areas.

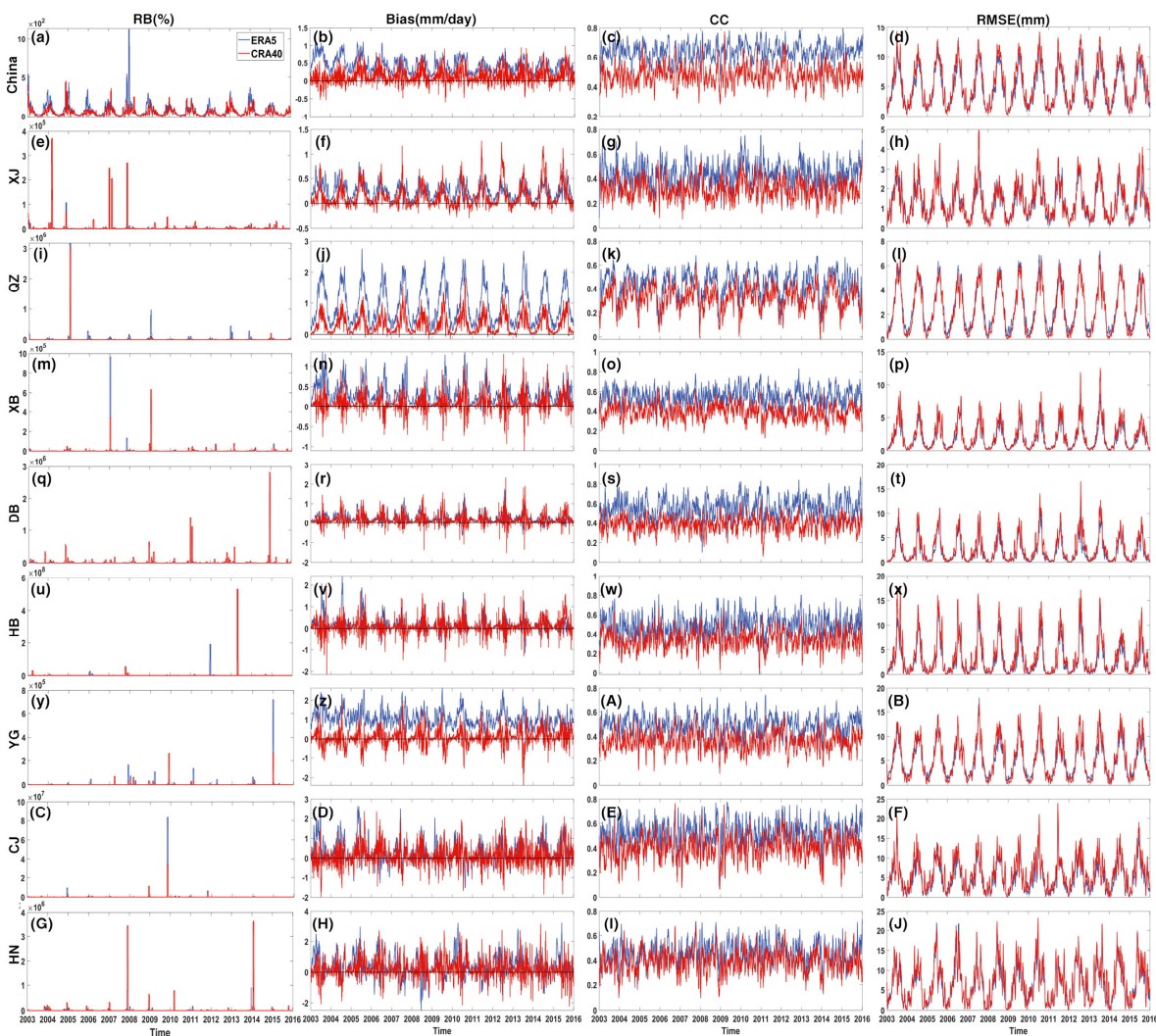

**Figure 6.** Daily variations of precipitation in terms of RB, bias, CC and RMSE for (**a–d**) China, (**e–h**) XJ, (**i–l**) QZ, (**m–p**) XB, (**q–t**) DB, (**u–x**) HB, (**y,z**), and (**A,B**) YG, (**C–F**) CJ, and (**G–J**) HN regions. The vertical axis scales for each subregion were adjusted based on the maximum values of the *y*-axis in different subpanels.

### 3.4. Probability Distributions by Occurrence and Precipitation Volume

The probability distribution function (PDF) reveals the inhomogeneity of precipitation in time and space, and has the ability to characterize precipitation products to detect precipitation. Precipitation in mainland China exhibits large variations in time and intensity. Figures 7 and 8 reveal the ability of the reanalysis precipitation products to detect different threshold precipitation rates in terms of daily-scale precipitation event occurrence (PDFc) and precipitation volume (PDFv), with PDFc as well as PDFv computed only in network cells with non-zero values of both reanalysis precipitation product and surface rain gauge precipitation observations [40]. It is noteworthy that the reanalysis precipitation products and the rain gauge observations exhibit similar trends, showing a higher occurrence of precipitation when the precipitation rate is below 5 mm/day. However, at precipitation rates below 1 mm/day (sporadic precipitation) and above 30 mm/day (heavy precipitation), the rain gauge observations show a higher frequency of precipitation events compared to both CRA40 and ERA5, suggesting an underestimation with the reanalysis products in

these ranges. Specifically, CRA40 precipitation occurrences align more closely with the rain gauge observations at rates below 1 mm/day, while ERA5 has precipitation occurrences more consistent with the rain gauge observations at rates above 30 mm/day. On the other hand, for precipitation rates ranging from 1 mm/day to 25 mm/day (light and medium precipitation), both CRA40 and ERA5 generally yield a higher number of precipitation events compared to the rain gauge observations. Moreover, ERA5 exhibits a higher incidence of precipitation compared to CRA40 within this range of 1–30 mm/day. This suggests that CRA40 performs better in capturing light and medium-weight precipitation but has limited capability in detecting heavy precipitation, while the opposite holds true for ERA5.

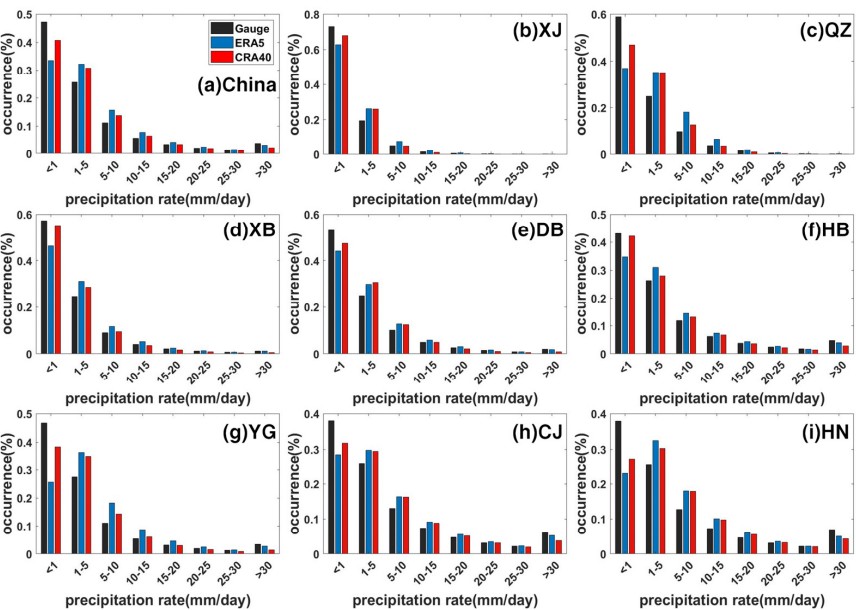

**Figure 7.** Characteristics of the probability distribution of precipitation occurrence (PDFo) in mainland China and eight subregions.

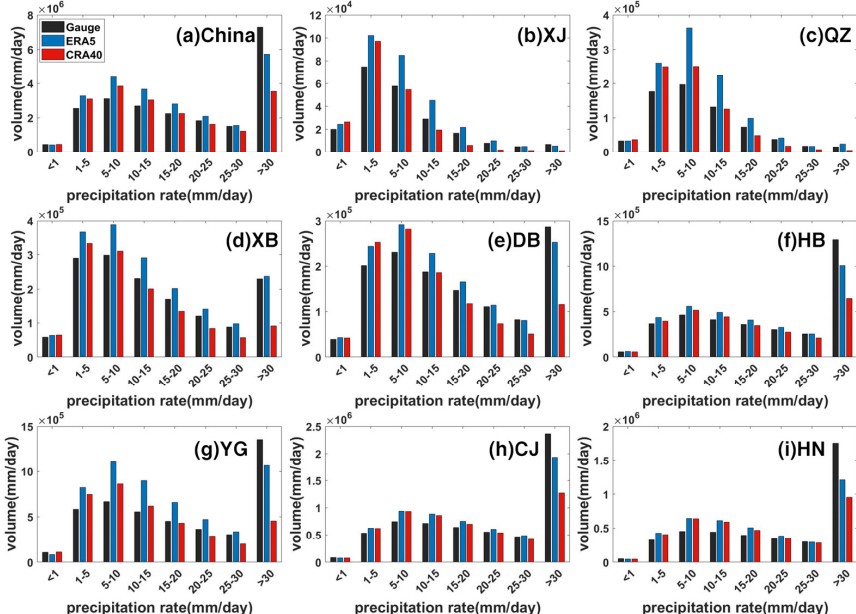

**Figure 8.** Characteristics of the probability distribution of precipitation volume (PDFv) in mainland China and eight subregions.

In addition, when precipitation rates are less than 1 mm/day, the precipitation amounts from the two reanalysis precipitation products are more consistent with the

rain gauge observations. At precipitation rates from 1 mm/day to 30 mm/day, ERA5 estimated more precipitation than rain gauges, and CRA40 performs relatively better than ERA5, which is consistent with the precipitation shown in Figure 2. However, at precipitation rates greater than 30 mm/day, these two reanalysis precipitation products estimate less precipitation than the rain gauges, except for areas XJ, QZ, and XB, where the ERA5 estimates precipitation more closely aligned with the gauge observations. This result suggests that both CRA40 and ERA5 underestimate the amount of precipitation for sporadic and heavy precipitation, and overestimate the amount of precipitation for light and medium precipitation, while CRA40 significantly underestimates the amount of precipitation for heavy precipitation. In summary, in the precipitation detection of China and its eight subregions, CRA40 has better precipitation detection performance when the precipitation rate is lower than 30 mm/day. When the precipitation rate is greater than 30 mm/day, the detection performance of ERA5 is better.

### 3.5. Daily Precipitation Detection Capability

Figure 9 shows the performance of CRA40 and ERA5 in terms of the detection rate (POD), critical success index (CSI), and false alarm rate (FAR) for China and eight subregions, with thresholds ranging from 1 to 250 mm/day at intervals of 1 mm/day. Both reanalysis precipitation products exhibit a general decreasing trend in POD and CSI as the thresholds increase, while the FAR displays an increasing trend. Overall, ERA5 outperforms CRA40 in terms of these statistical metrics. Specifically, ERA5 demonstrates a higher POD, higher CSI, and lower FAR for most threshold intervals in mainland China, except for the precipitation rates in the range of 50–150 mm/day, where CRA40 achieves a slightly better POD than ERA5. In the other eight subregions of China, ERA5 consistently shows a higher POD, CSI, and lower FAR compared to CRA40. It is worth noting that both CRA40 and ERA5 exhibit low PODs and CSIs (close to 0%) and high FARs (close to 100%) near the threshold value of 250 mm/day. This indicates that neither ERA5 nor CRA40 perform well in accurately detecting and capturing precipitation events at high precipitation rates. In summary, ERA5 generally surpasses CRA40 in terms of the POD, CSI, and FAR, indicating its superior performance in detecting precipitation events across various threshold values. However, both CRA40 and ERA5 exhibit limitations in accurately capturing high precipitation rates.

### 3.6. Spatial Analysis

Figure 10 shows the spatial distribution of bias, RB, RMSE and CCs for the CRA40 and ERA5 reanalysis precipitation products for the 13-year daily mean precipitation in mainland China, which will help hydrological modelers to analyze the error propagation during hydrological simulation. As can be seen in Figure 10, the spatial distribution of precipitation bias, RB, RMSE, and CCs in mainland China exhibit similar patterns for CRA40 and ERA5. These two datasets generally overestimate precipitation in the northern and western regions of China, with ERA5 showing a greater degree of overestimation. These two datasets slightly underestimate precipitation in the southeast. Notably, ERA5 significantly overestimates precipitation in the YG region, while CRA40 exhibits comparatively less overestimation. The RMSE values are generally close to 0 mm in most areas. Higher RMSE values are primarily found in humid climate regions (HN, CJ, and YG) and arid climate regions (XJ). Importantly, the RMSE of ERA5 in the southeastern part of the QZ region is notably higher than that of CRA40. Regarding the CC, CRA40 performs better than ERA5. Higher CC values are observed in eastern, southeastern, and northwestern China, ranging mostly between 0.7 and 1. Slightly lower CC values are seen in the YG and northwestern QZ regions. Both CRA40 and ERA5 generally exhibit positive precipitation biases, particularly in the YG, HN, and CJ regions, with ERA5 displaying a more pronounced bias. In contrast, positive biases are relatively small in the XJ, XB, DB, and HB regions. It is noteworthy that CRA40 demonstrates a significantly smaller bias in the QZ region, whereas ERA5 exhibits a substantial positive bias.

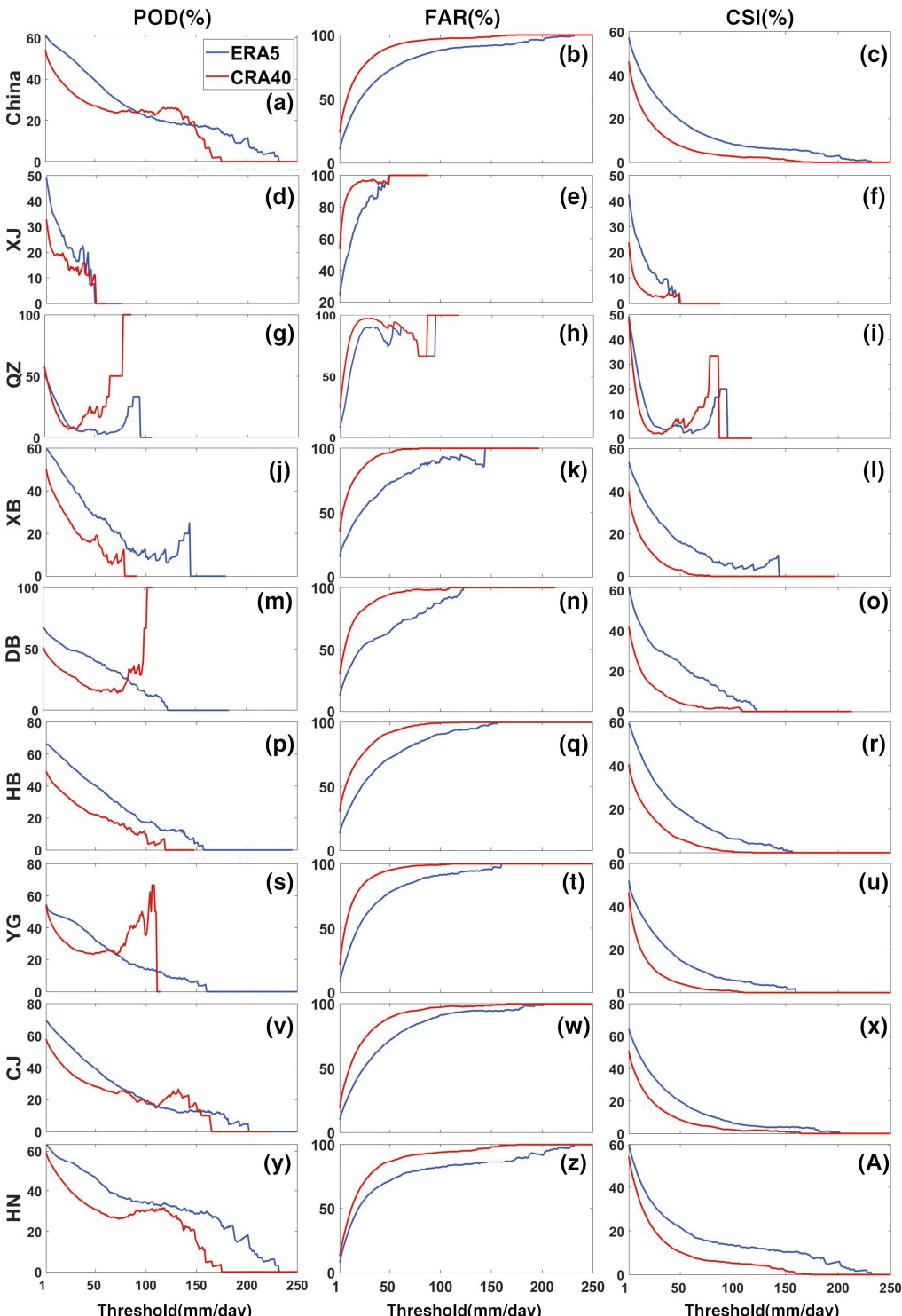

**Figure 9.** Contingency metrics for precipitation POD, FAR, and CSI in (**a–c**) China, (**d–f**) XJ, (**g–i**) QZ, (**j–l**) XB, (**m–o**) DB, (**p–r**) HB, (**s–u**) YG, (**v–x**) CJ and (**y,z,A**) HN regions. The vertical axis coordinate scale is set according to the maximum value of the *y*-axis in different subpanels.

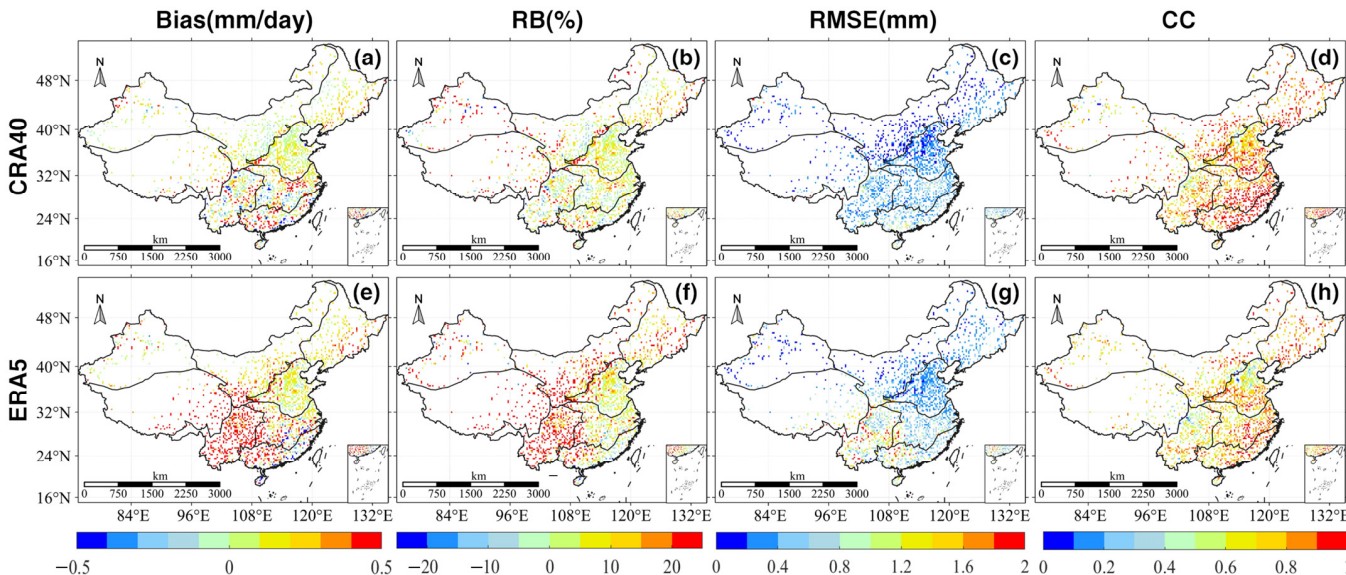

**Figure 10.** Spatial distribution of bias, RB, RMSE, and CC for (**a**–**d**) CRA40 and (**e**–**h**) ERA5 in China.

Figure 11 shows the occurrence probability and occurrence cumulative distribution functions of the bias, RB, RMSE, and CC statistics within the Chinese region. As can be seen in Figure 11, there is a significant difference between CRA40 and ERA5 in PDFc and CDFc for the deviation, RB, RMSE, and CC statistics, and CRA40 performs relatively better. The PDFc and CDFc for CRA40 indicate that there are significantly more pixels, with an estimated bias equal to 0 mm/day, than there are for ERA5. Similar conclusions can be drawn in the RMSE values in the range of 0–0.4 mm. Furthermore, in order to better compare the performance of the two reanalysis precipitation products, a set of statistics used to evaluate the quantification is given below. For CRA40 and ERA5, 24.1% and 51.23% of the deviations in the 13-year daily average precipitation are greater than 0.5 mm, 63.38% and 82.10% of the RB are greater than 5%, 47.59% and 74% of the RMSE are greater than 0.4 mm, and 88.36% and 73.90% of the CC are greater than 0.7, respectively.

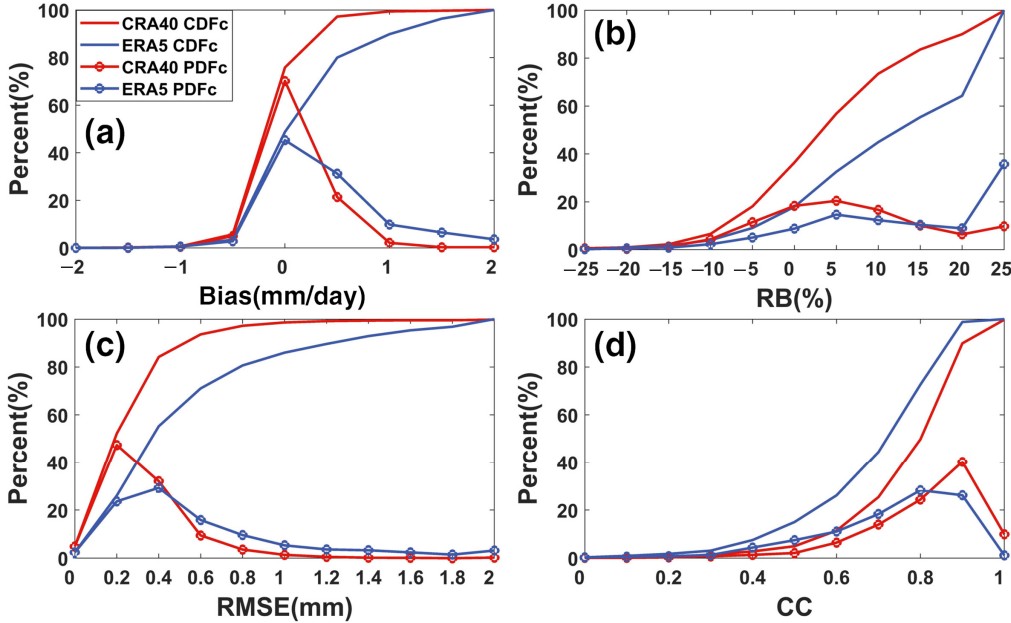

**Figure 11.** Probability (cumulative) distribution function PDFc (CDFc) for (**a**) bias, (**b**) RB, (**c**) RMSE, and (**d**) CC values based on the results calculated in Figure 10.

## 4. Discussion

This study uses the eight indicators described above to assess the 13-year daily mean precipitation, seasonal daily mean precipitation, and daily precipitation at different time scales over the period of 2003–2018. We find some bias and different regional applicability of the CRA40 and ERA5 reanalysis precipitation products to rain gauge observations within mainland China. The vast expanse of China results in distinct regional variations in precipitation patterns, and these localized differences are quite noticeable. The accuracy of precipitation forecasts is significantly influenced by the intensity of rainfall in a particular area [41,42]. Therefore, to further explore the characteristics and applicability of CRA40 and ERA5 under different intensities of precipitation in mainland China and its subregions, precipitation was classified into six classes according to the Chinese national standard Classification of Precipitation Classes (GB/T 28592-2012) [38] and analyzed using probability distributions based on the number of occurrences and amount of precipitation.

As shown in Figure 2, it can be seen that both CRA40 and ERA5 can better simulate the spatial variation trend of precipitation in China, but the spatial distribution characteristics of precipitation in the CRA40 product are more similar to those of the GGKRIG. It is worth noting that ERA5 and CRA40 may have significantly overestimated the annual mean daily precipitation in the southern Himalayas in the YG region. This overestimation could be attributed to the absence of surface rainfall gauge stations in that area, resulting in the underestimation of precipitation due to the influence of surrounding rain gauge station values during the interpolation process.

As can be seen from Figure 3, CRA40 (ERA5) has a small number of significantly underestimated (overestimated) data points. After a careful check of the locations of the underestimated and overestimated points in the above regions, it is found that the underestimated points of CRA40 are distributed at the sea–land junction of the Yangtze River estuary. This is probably attributed to the impact of the missing values at the sea–land boundary. As for ERA5, and the overestimated points are distributed at the mountainous areas and mountain ranges with complex topography in parts of YG and HN (Figure 2c,d). These results are consistent with those of Fallah et al. [43] and Nogueira et al. [44], whose findings suggest that ERA5 has poor-quality precipitation estimation in mountainous and mountain regions with complex precipitation processes. Based on the index calculation results in Table 3, it can be seen that CRA40 has relatively better applicability in Chinese mainland and seven subregions except XJ, and the product quality of ERA5 in the XJ region is slightly better than CRA40.

The daily bias and RMSE generally exhibit annual periodicity and some correlation, with the highest RMSE occurring in summer and the lowest in winter. This pattern is particularly pronounced in the XJ and QZ regions. In addition, the cyclical characteristics of bias and RMSE are relatively more pronounced in the QZ and XJ regions, suggesting that the two reanalysis precipitation products are more influenced by seasonal climate and surface features. In the CJ and HN regions, both the CRA40 and ERA5 products exhibit more negative deviations. This is likely attributed to the complex topography of these regions, including mountain ranges and hills, which affect precipitation distribution. Additionally, the presence of extensive vegetation and wetlands in these regions, along with their status as rainy seasons, further complicates the estimation and observation of precipitation.

In the daily precipitation detectability assessment, both CRA40 and ERA5 are highly biased relative to the rain gauge observations for light rainfall (1–10 mm/day), while the opposite is true for heavy precipitation (greater than 30 mm/day), which is consistent with the findings of Hénin et al. [45] and Sharifi et al. [46]. The poor detection of heavy precipitation may be due to the fact that the reanalysis precipitation products are assimilated or simulated from multiple observations. And the accuracy of the original input data and the accuracy of the simulation algorithms will affect the accuracy of the reanalysis precipitation products [47]. China is a vast country with complex geographical units and diverse climatic environments, and ground stations are sparse in some parts of the country,

making input data from satellites and radar more uncertain. In addition, both CRA40 and ERA5 reflect only the average state of precipitation over a $0.25° \times 0.25°$ grid area, with limited ability to detect localized precipitation in complex terrain environments. At the same time, the regional representativeness of the rain gauge observations is low (especially in the QZ and XJ regions), which in turn leads to a certain bias between the reanalysis precipitation products and the ground observations.

## 5. Summary

The applicability of the first generation of the CRA40 daily precipitation product to mainland China has been assessed in terms of both categorical and continuous statistical indicator analyses, with the daily-scale rain gauge observations from 2003 to 2015 from the China Meteorological Information Center (CMIC) as a reference, and the ERA5 daily precipitation product as a comparative assessment. The main findings are summarized as follows:

(1) CRA40 performs better than ERA5 across mainland China in terms of the 13-year daily average precipitation. Compared to ERA5, CRA40 exhibits a higher CC (0.97), a smaller RB (5.25%), a lower RMSE (0.34 mm), and a smaller FSE (0.05), while ERA5 has a lower CC (0.91), a larger RB (18.59%), a slightly higher RMSE (0.75 mm), and a higher FSE (0.24). Both CRA40 and ERA5 show less overestimation of precipitation in wet regions (CJ and HN), but exhibit more pronounced overestimation in high-altitude and dry climatic regions (QZ and XJ). Additionally, CRA40 generally had smaller RB and RMSE values in wetter regions and higher CC values compared to ERA5, except in XJ, where the CC values are slightly lower than ERA5.

(2) Seasonally, CRA40 has less overestimation and higher CC in areas with abundant precipitation over southeastern China (HN, CJ, and HB). The RB values of ERA5 are relatively large in all seasons (22.93%, 12.81%, 20.32%, and 36.54%), while CRA40 exhibits only slight overestimation in all seasons (6.89%, 4.77%, 3.74%, 6.86%). Additionally, CRA40 had high CC values ($\geq 0.93$), while ERA5 had lower CC values ($\geq 0.84$).

(3) ERA5 precipitation products have better quality than CRA40, and are more suitable for daily-scale precipitation studies in mainland China and its sub-divisions. Although the daily bias of ERA5 is relatively large, it has higher daily series CC values, which can better reflect the characteristics and variations in precipitation events and provide a reliable basis for the assessment of precipitation (Figure 6).

(4) The analysis of probability density functions for sporadic precipitation (<1 mm/day) and light to medium rainfall (1–25 mm/day) shows that CRA40 better captured these categories, whereas ERA5 performs better in capturing heavy precipitation (>30 mm/day). Both CRA40 and ERA5 underestimate trace and heavy rainfall and overestimate light and moderate rainfall, with CRA40 underestimating heavy precipitation to a greater extent.

(5) ERA5 exhibits better contingency statistics than CRA40, with a higher POD and CIS and a lower FAR for most threshold intervals. Both CRA40 and ERA5 have poor performance in detecting high precipitation rates.

(6) Both CRA40 and ERA5 exhibit better performance in estimating precipitation during the spring, summer, and autumn seasons compared to winter. For most of the time, both CRA40 and ERA5 demonstrate a pronounced overestimation, particularly in the QZ region at higher altitudes. CRA40 shows a lower level of overestimation (51.62%) during winter in comparison to ERA5 (299.84%), but its CC (0.36) is lower than that of ERA5 (0.76) (Figures 4 and 5 and Table 3).

(7) In the high-altitude QZ region, the quality of CRA40 precipitation products is poor in winter, showing a low correlation, while CRA40's product quality is better than ERA5 in the other three seasons. In the arid area of XJ with sparse precipitation, ERA5 is more suitable than CRA40. Although ERA5 has a relatively large relative deviation, it generally has a higher correlation. In the YG, CJ, and HN regions, where precipitation is abundant and the terrain is complex, CRA40 has better application

potential than ERA5. In the XB, DB, and HB regions with less precipitation, except for the high deviation and relatively low correlation of ERA5 in winter in the XB region, the performance of CRA40 and ERA5 products is relatively stable and reliable (Table 3).

This study identifies and quantifies the error characteristics of the CRA40 and ERA5 reanalysis daily precipitation products, which are very important for hydrological applications in mainland China. The results of the performance evaluation study on the first-generation global land surface reanalysis daily precipitation product, CRA40, and the fifth-generation European reanalysis product, ERA5, in China is expected to provide credibility to the use of the two products and scientific references in hydrological applications. In mainland China, CRA40 has better precipitation observation accuracy than ERA5 for 13-year daily average precipitation and seasonal daily average precipitation, which indicates that the CRA40 daily precipitation product has high potential for applications in precipitation research and hydrological simulation in mainland China. However, there is still a gap between CRA40 and ERA5 in terms of continuous daily precipitation observation accuracy and error. In addition, the relative bias between CRA40 and ERA5 is large in the QZ region in winter, which can be calibrated through ground observations to improve the accuracy in the subsequent improvement of the product [48,49]. Although reanalysis products have their limitations, they still have important value in studying climate change and exploring historical weather events. When using CRA40 and ERA5 to reanalyze precipitation products, their characteristics and uncertainties should be fully considered, and comprehensive analysis and judgment should be conducted in conjunction with other observational data.

**Author Contributions:** Conceptualization, Z.Z. and S.C.; methodology, Z.Z. and S.C.; validation, S.C.; formal analysis, S.C. and Z.L.; investigation, S.C., Z.L. and Y.L.; resources, S.C. and Z.L.; data curation, Z.Z. and S.C.; writing—original draft preparation, Z.Z.; writing—review and editing, S.C. and Z.L.; visualization, Z.Z.; supervision, S.C. and Z.L.; project administration, S.C., Z.L. and Y.L.; funding acquisition, S.C. and Z.L. All authors have read and agreed to the published version of the manuscript.

**Funding:** This research was funded by Guangxi Key R&D Program (Grant No. AB22080104, AB22035016); Guangxi Natural Science Foundation (2020GXNSFAA238046); Key Laboratory of Environment Change and Resources Use in Beibu Gulf (Grant No. NNNU-KLOP-K2103) at Nanning Normal University; and Innovation Group Project of Southern Marine Science and Engineering Guangdong Laboratory (Zhuhai) (No. 311022001).

**Data Availability Statement:** The daily rain gauge observations used in this study can be downloaded from the official website of China Meteorological Administration (https://data.cma.cn/, accessed on 10 December 2021). In addition, the reanalysis precipitation data used in this study are the fifth generation of the European reanalysis product, ERA5 (https://cds.climate.copernicus.eu/#!/home, accessed on 7 October 2022), and the first generation of China's global land surface reanalysis daily precipitation product, CRA40 (https://data.cma.cn/, accessed on 7 October 2022).

**Acknowledgments:** Thanks are given to Qin Jiang from East China Normal University and Huiqin Zhu from Nanning Normal University for their helpful advice on data processing. We also thank the three reviewers for their detailed review and helpful comments and suggestions.

**Conflicts of Interest:** The authors declare no conflict of interest.

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
