# Peer review of "An Evaluation of CRA40 and ERA5 Precipitation Products over China"

_remotesensing, doi:10.3390/rs15225300_

Round 1

Reviewer 1 Report

Comments and Suggestions for Authors

This article evaluate the performance of two reanalysis precipitation products over mainland China, including the first-generation Chinese global land-surface reanalysis precipitation product (CRA40) and the fifth-generation European reanalysis precipitation product (ERA5). The study redults is important for the improvement of reanalysis precipitation products. However, the readability of this article might be improved if some of the following aspects could be explained.

1. Please Check the maximum number of subregions mentioned on line 177.

2. As mentioned by the authors on line 208, precipitation product of CRA40 includes precipitation and snow. Does that mean the precipitation of CRA40 is the sum of rainfall and snowfall. Then please indicate whether the precipitation product of ERA5 and ground stations also represent the sum of rainfall and snowfall.

3. I can't find the red letters A and B mentioned on line 216 in Figure 2d

4. Please check the range of precipitation intensity represented by the sixth level of precipitation in Table 2.

5. Please indicate the meaning of the white area in Figure 2d.

6. Table 3 mentioned on line 365 can't be found.

7. I would like to ask the authors to label or indicate the season that each image represents in Figure 5.

8. It is hoped that the time series changes of RB in different regions can be shown simultaneously in Figure 6.

9. Only a few references have been included in the manuscript. Please add relevant references to enhance the global importance of the paper.

10. The authors evaluate the accuracy of precipitation products in multiple regions. Hope to emphasize more regional differences in the quality of precipitation products in the conclusion or abstract.

Overall, the muniscript is possible to be published after fully consideration of aforementioned issues.

Reviewer 2 Report

Comments and Suggestions for Authors

I wrote comments in the document. 

Reviewer 3 Report

Comments and Suggestions for Authors

This paper aims to evaluate the performance of two distinct reanalysis precipitation products, CRA40 and ERA5. This comparison and evaluation is important for the community in deciding the use of different products over China. The paper is well organized and the figures and conclusions are clearly stated. 

Comments on the Quality of English Language

The quality of English does need to be improved. 
